# Environmental Quality Perceptions and Health: A Cross-Sectional Study of Citizens of Kaunas, Lithuania

**DOI:** 10.3390/ijerph17124420

**Published:** 2020-06-19

**Authors:** Regina Gražulevičienė, Sandra Andrušaitytė, Audrius Dėdelė, Tomas Gražulevičius, Leonas Valius, Violeta Kapustinskienė, Inga Bendokienė

**Affiliations:** 1Department of Environmental Science, Vytautas Magnus University, 44248 Kaunas, Lithuania; sandra.andrusaityte@vdu.lt (S.A.); audrius.dedele@vdu.lt (A.D.); t.grazulevicius@gmail.com (T.G.); 2Department of Family Medicine, Lithuanian University of Health Sciences, 48005 Kaunas, Lithuania; Leonas.Valius@lsmuni.lt (L.V.); Violeta.Kapustinskiene@lsmuni.lt (V.K.); 3Division of City Planning and Architecture, Kaunas City Municipality, 48005 Kaunas, Lithuania; inga.bendokiene@kaunas.lt

**Keywords:** citizen science, environmental health, neighborhood perceptions, hypertension, physical activity

## Abstract

The perception of urban environmental quality is an important contributor when identifying local problems in sustainable development and environmental planning policy. This study examined the associations between environmental and social residential characteristics, physical activity, obesity, and hypertension in Kaunas city, Lithuania. This cross-sectional study analyzed 580 citizens’ demographic-, socioeconomic-, health-, and lifestyle-related factors, environmental health concerns, and environmental quality perceptions. Using Geographic Information Systems and the multivariate logistic regression, we found that the less physically active group more often presented lower than mean ratings of the quality of pathways and cycling routes (32.9% and 45.6%, *p* = 0.042) and only irregularly visited the natural environment. Obese participants presented poorer ratings of air pollution, the quality of pathways and cycling routes, their possibility to reach green spaces by walking, and the available relaxing areas. The environmental issues associated with hypertension were poor possibilities to reach green spaces by walking (OR 1.94, 95% CI 1.14–3.32) and the availability of relaxation areas (OR 2.30, 95% 1.34–3.95). The quality of the neighborhood and individual-level characteristics were the factors that influenced a higher prevalence of health problems at the district level. Our findings suggest that a public health policy to improve the physical and social environment of the neighborhood would have a potential to increase citizens’ physical activity and health.

## 1. Introduction

Public health specialists have been increasingly interested in how the quality of citizens’ residence settings is associated with physical activity and health [1,2]. Physical inactivity is a major modifiable risk factor for many preventable chronic non-communicable diseases, and there is a potential risk threshold for health related to the degree of activity or inactivity [3]. The physical environment, residential availability of natural outdoor environments, and accessibility and aesthetics are important factors that influence physical activity [2,4]. The researchers who investigate possibilities for the promotion of health and physical activity among citizens emphasize the significance of the citizens’ perceptions of the neighborhood’s quality in facilitating physical activity [5,6].

The growing awareness of the importance of the environmental health and social factors for human health has led to the participant-approach studies that might more adequately reflect the local environmental issues, overall health, and the wellbeing of citizens. Citizen science in public health might benefit from the citizens’ active contribution with their intellectual effort and surrounding knowledge [7,8]. However, to date, citizen science has not relied heavily on the evidence of the environmental impact on citizens’ health, and there are opportunities to engage those affected by environment-related health problems, such as cardiovascular disease, hypertension, allergies, and other chronic diseases.

Epidemiological studies have shown that contact with green space has beneficial effects such as reduced risks of cardiovascular disease, obesity, or diabetes, and lower risks of all-cause mortality [9,10]. There is growing scientific evidence that both physical activity and contact with urban green spaces have the potential to contribute positively to citizens’ health [11,12,13]. Meanwhile, urban planning, city transport system, and neighborhood characteristics may have a significant negative influence on human health and well-being by affecting the environmental quality, or, indirectly, by influencing behaviors such as physical activity—which, in turn, affects chronic disease [14]. Given the complexity of the built environment, understanding its influence on human health requires a community-based, multilevel, interdisciplinary research approach [15]. Multi-level and urban system approaches have the potential of having larger-scale effects and should be evaluated [16].

To date, little is known about how people’s perceptions of the quality of residential settings are related to their physical activity and health concerns. The idea of this article is to assess the associations between the ratings of various environmental (built and social) residential characteristics and health indices (blood pressure, obesity, and hypertension), which have numerous common individual and environmental risk factors. The results of this study might present possible target areas for interventions in physical activity, including the environmental, individual, social, and policy levels.

The present research reports on the first-year findings from a Kaunas pilot study, which is a part of the Horizon 2020 proposal Citizen Science for Urban Environment and Health (CitieS-Health, *http://citieshealth.eu/*) and has been organized by Vytautas Magnus University, Kaunas, Lithuania. Using environmental epidemiological approach, we seek to answer the study participants’ question: “Why do citizens in my district suffer from hypertension more often than in other ones?” To answer the question, we formulated three objectives: 1) To examine citizens’ concerns and perceptions of the quality of their neighborhoods; 2) to explore the relationships between individual- and area-level characteristics and the prevalence of hypertension; 3) and to test the hypothesis about the rating of the relationship between the neighborhood quality and the citizens’ physical activity, obesity, and diastolic blood pressure.

## 2. Materials and Methods 

### 2.1. Study Design

The pilot study of the Urban Environment and Health proposal was conducted between July and February of 2020. We conducted a cross-sectional study among 580 adults who lived in Kaunas city, Lithuania. Our main engagement methods included radio information, announcements in local newspapers and web sites, advertisements at community events, and conferences. Because the citizens’ response to the invitation via mass media was poor, we also used the list of the participants of the scientific-practical conference Human and Nature Safety to send personal invitations to join the study. From the beginning of the project, the participants were involved in discussions to identify and prioritize environmental and health issues. The three major environmental concerns the citizens expressed are as follows: air pollution, traffic noise, and the availability of cycling and smooth walking paths. The major health concerns were hypertension, obesity, and cardiac problems. During the first year of the project, the participants together with scientists were involved in identifying the relevant questions, planning the design of the Kaunas pilot study, and discussing the study questionnaire and protocol. Using the interviews, we identified citizens’ environmental concerns and major health concerns and formulated the aim of the study. The participants of the study were volunteers who filled out a questionnaire that had closed-ended and open-ended questions and signet the Informed Consent form. In the questionnaire, we asked the participants about the characteristics of the built and the social residential environment, the duration of living in the current place of residence, health behavior, and chronic or other diseases diagnosed by physicians. In addition, the participants were asked to take part in seven-day physical activity measurements with physical activity sensors FitBit Alta. About a half of all the participants agreed to take part in these measurements, which were conducted during the second year of the project.

The participants filled out the questionnaire during face-to-face interviews. We used a seven-point Likert rating scale to measure environmental perceptions, and then analyzed the associations with the self-reported health status. The Likert scale included a series of questions to be answered and 7 balanced responses the participants could choose from. Based on survey, and the discussions between the participants’ and scientists the following research question was formulated: “Why do citizens in my district suffer from hypertension more often than in other ones?” We conducted geospatial analysis using the ArcGIS mapping (Esri, Redlands, CA, USA) and analytics platform and statistical analysis using the SPSS version 25.0 package (IBM Corporation, New York, NY, USA) to assess the associations between the perception of the environmental quality, self-reported physical activity, and the participants’ health and well-being.

### 2.2. Ethics Statement

Kaunas Regional Committee for Biomedical Research Ethics approved the study protocol, the questionnaire, the Subject Information Form, and the Informed Consent Form as well as the consent procedure (BE-2-51. 2019-06-10). We informed the participants about the study in detail, and the participants signet the Informed Consent Form. At the beginning of the study, all the collected personal data were recorded on “paper” documents specifically designed for biomedical research, and then were given numerical values for anonymous storage in electronic media. The personal data of the participants were stored in accordance with the “Legal Protection of Personal Data” act. All personal data collected were coded by giving them numerical values. The coded anonymous data were stored separately from the personal identification data. Only depersonalized individual data were used for data analysis. Biomedical examination was conducted in accordance with the Declaration of Helsinki and other relevant regulatory requirements.

### 2.3. Analysis

We used the addresses provided by the participants to create variables with the residence district/community categories. We evaluated exposure to major traffic flows (more than 10,000 cars per day) by the geographic information system (GIS). The participants’ perception on the characteristics of the quality of the built environment were assessed using questions on the infrastructure in the residence neighborhood, public transport, pathways and cycling routes, walking distances to the city’s green spaces or parks, areas adapted to exercise and relaxation, public spaces to meet people, and neighborhood safety. In addition, we asked questions on problems caused by air pollution and noise in the place of residence, and on the average time per day spent outdoors. We assessed the different environmental variables by using a seven-point Likert rating scale ranging from 1 (strongly disagree) to 7 (strongly agree) in order to measure mean environmental perceptions. The self-reported health was evaluated using a five-point Likert rating scale ranging from 1 (great) to 5 (poor). Higher scores indicated better neighborhood conditions.

The individual socio-economic status (SES) was assessed by evaluating the respondents’ income, educational status, and situation at work. We assessed the participants’ health status by the presence or absence of physician-diagnosed chronic diseases, systolic and diastolic blood pressure, and body mass index (BMI) calculated using the measures of body weight and body height. We asked the participants about smoking history and the average time per day spent outdoors. We dichotomized personal data and used mean values of environmental perception scores as cut points.

Information on hypertension was obtained by asking the participants to answer the question “Have you been diagnosed with hypertension?” and/or asking them if their systolic blood pressure was 140 mmHg or higher and/or their diastolic blood pressure was 90 mmHg or higher. These health measures were dichotomized in the analysis in order to receive a binary outcome variable for an easier interpretation of logistic regression estimates. The duration of physical activity was evaluated by the question “During the last week, what was the mean time per day you spent outdoors by fast walking, biking or gardening?” The participants were then classified into two groups according to the Public Health Guidelines for Physical Activity [17]—at least 150 min/week of moderate-intensity physical activity outdoors recommended duration or fewer min/week spent outdoors. We used the participants’ height and weight measures to calculate the body mass index (BMI) (kg/m^2^). The participants with the BMI above 30 were classified as obese.

Individual-level predictors of the socio-economic status (SES) were education level (unfinished secondary school/finished secondary school but below BA (Bachelor’s degree)/BA or higher), and the type of occupation (worker, student, unemployed—low; housekeeper, officer—medium; manager, company owner—high socio-economic status).

We used the chi-squared test to compare the values and the frequencies of baseline characteristics at the individual and the neighborhood levels. Statistical significance was set at *p*-value < 0.05. We used data on the traffic flows in the residence street to classify the participants’ levels of exposure to traffic emissions. The participants residing in the street with above 10,000 cars/day we classified as exposed to heavy traffic emissions in their place of residence. We compared the perceptions and health concerns in the districts with a higher prevalence of hypertension (presented by >mean) and the districts with a lower prevalence of hypertension (presented by <mean), and evaluated the associations as odds ratios (OR) and their 95% confidence intervals (CI) controlling for covariates that were known to be related to the health status. Quantitative variables were reported as mean values and standard deviations. We evaluated the associations between the covariates that were related to walking and cycling (the quality of routes) and physical activity.

The statistical literature recommends applying higher than 0.05 *p*-value thresholds (such as 0.2) for the inclusion of predictors from bivariate statistics, in order to prevent the exclusion of relevant factors [18,19]. For this reason, the predictor variables whose univariate tests showed an association of *p* < 0.2 with the outcome or those that changed the adjusted odds ratios (aOR) by 10% or more were retained for inclusion in the multivariate logistic regression analysis. We used Fisher’s exact tests to compare the qualitative characteristics between the groups. A multivariate logistic model was used to explore the independent associations between the health outcomes and environmental exposures, controlling for sex, education level, age, smoking status, and income. In addition, we evaluated the associations of prevalent hypertension with the presence or absence of a major road in the residence area and the perception of air pollution among all participants. Significance was accepted at an alpha level of 0.05. Statistical analyses were performed using SPSS version 25.0.

## 3. Results

### 3.1. Respondents’ Characteristics

The participants of this study were 576 citizens; the majority of them were of working age, about 61.7% of the participants were of 18–44 years age, and 7.9% of them were of retirement age (≥ 65 years). As much as 44% of all the participants had university degrees. As shown in Table 1, the majority of the participants (57.9%) were female, 16.8% of the participants had monthly net income of less than 400 Euro, and about 22% of the participants had been exposed to heavy traffic emissions in their place of residence for 18 years or more.

We used the map of Kaunas city transport network to form an annular-radial structure. We explored spatial patterning in air pollution perceptions (presented by the <mean or >mean score) and the prevalence of hypertension (in percent) at the district level (Figure 1). In different city districts, the prevalence of hypertension exhibited different spatial patterns—it ranged from 15.0% to 51.3%. Citizens in the districts of the central part of the city were more concerned about the health consequences of poor air quality, while participants residing in the periphery were concerned less. In the districts of high perception of air pollution (Eiguliai, Petrašiunai, and Šančiai—score above the mean), higher than mean (33.3%) prevalence of hypertension predominated. A heavy-traffic street network covers these districts. In the districts located on the hills (Žaliakalnio and Dainavos) citizens were less concerned about the health consequences of poor air quality, and the prevalence of hypertension was lower. Pearson’s correlation analysis showed no association between the scores of the perception of the air pollution and the prevalence of hypertension at the city level (r = −0.032, *p* = 0.497). However, in Eiguliai and Šančiai districts, we found a negative correlation. The correlation was weak, but significant, −0.292 (*p* = 0.021) and −0.351 (*p* = 0.036), correspondingly. This association was not adjusted for possible confounding variables such as: age, sex, education level, smoking, or income. 

### 3.2. Health Concerns and the Perceptions of Neighborhood Quality

As much as 46.7% of the participants suffered from one or more chronic diseases. Hypertension, obesity, and cardiovascular diseases were mostly common in participants over 45 years of age. To find the answer to the question that interested the participants the most—“Why do citizens in my district suffer from hypertension more often than in other ones?”—we calculated the prevalence of hypertension in different districts. In the city, the mean prevalence of hypertension was 33.3%. The districts with the prevalence of hypertension above the mean were classified as high-prevalence districts. We compared physical activity and health measures in district groups by the prevalence of hypertension (Table 2). The participants who lived in the districts with a higher prevalence of hypertension more often had a higher systolic blood pressure, a greater body mass index, and obesity, compared to those living in the districts with a low prevalence of hypertension. Chronic diseases and self-reported “poor” health status were more common in participants who lived in district groups with a high prevalence of hypertension.

On weekdays, self-reported physical activity was low in both groups, mostly not reaching the overall recommended duration of physical activity (at least 150 min/week of moderate-intensity physical activity outdoors). Only 11.8% of all the participants reached the recommended level physical activity. The evaluation of the mean duration of physical activity in a park per day during the previous showed no significant difference between the participant groups: the reported mean time in low-prevalence hypertension and high-prevalence hypertension districts was, correspondingly, 22.95 min and 24.13 min (*p* = 0.678).

In order to reveal the specific characteristics of the physically active participants, we compared the group of the participants who reached the recommended overall physical activity level (at least 150 min/week of moderate-intensity physical activity outdoors) with those who did not. We found that in both groups, the perceptions of the quality of the neighborhood were similar. However, the less physically active group more often presented lower than mean ratings of the quality of pathways and cycling routes (32.9% and 45.6%, *p* = 0.042). In addition, 54.9% of them only irregularly visited the natural environment, compared to 33.8% of more physically active participants (*p* = 0.001). In the group of participants who reached the recommended physical activity level, mean age, sex, education, and other personal-level data did not differ statistically significantly from those in the less physically active group.

Table 3 shows the participants’ perceptions of their neighborhood quality in the districts with low and high prevalence of hypertension. The lasts three questions of Table 3 present the participants’ social wellbeing (feeling of safety, significance, and stress level) in their place of residence. The answers to these questions indirectly show the quality of the social environment in the districts. The rating was mainly positive and similar in both groups of participants.

The participants in the districts with low and high prevalence of hypertension highly rated the statement that public transport in the district met their needs (5.39 and 5.20, accordingly) and recognized that there were good opportunities for walking to reach the city’s green spaces or parks (5.07 and 5.16, accordingly). The perception of air pollution in the place of residence as a cause of health problems was rated the lowest, showing a potential for improving the situation, and there were significant differences between the two groups: in the districts with a low prevalence of hypertension, it was 4.03, while in the districts with a high prevalence of hypertension, it was 3.62, *p* = 0.027. The participants’ satisfaction with pathways, cycling routes, and areas adapted to exercise and relaxation as well as the sense of security in their residence area were also rated lower in the districts with a high prevalence of hypertension, yet the differences between the groups were not statistically significant.

### 3.3. Associations between Individual- and District-Level Perception of Environmental Quality and Health Issues

We used three multivariate logistic regression models to examine the associations between neighborhood quality and physical activity, the obesity, and diastolic blood pressure, considering both individual-level and neighborhood-level variables. 

Table 4 presents odds ratios (OR) and 95% confidence intervals (CI) from the multivariate logistic regression models examining covariates and associations. The first model included physical activity and some environmental-related perceptions. The second one included obesity, and the third model included diastolic blood pressure. The data of all the models were adjusted for sex, education level, age, smoking status, and income.

The first model showed significant relationships between physical activity and the rating of the quality of pathways and cycling routes. More physically active participants presented poorer evaluations of the quality of pathways and cycling routes than less physically active participants did (OR 1.70 (95% CI 1.02–2.85)); in addition, they significantly more often visited green spaces regularly (OR 0.40 (95% CI 0.24–0.70)), and gave higher scores to the noise in their residence place (OR 0.37 (95% CI 0.20–0.67)).

Less physically active participants more often were obese and had a higher diastolic blood pressure.

The obese participants more often were male, older, and less educated. Obese participants presented poorer ratings of the quality of pathways and cycling routes (OR 1.95 (95% CI 1.17–3.27)), their possibility to reach green spaces by walking (OR 1.64 (95% CI 0.98–2.75)), and the available relaxation areas (OR 2.37 (95% CI 1.39–4.02)). Obese participants also presented poorer ratings of air pollution problems (OR 1.87 (95% CI 1.11–3.15)), and their places of residence more often were exposed to heavy traffic (OR 0.39 (95% CI 0.19–0.82)).

Diastolic blood pressure of 90 mm Hg or higher was more prevalent among older men and less educated participants. Hypertensive participants more often than participants with lower blood pressure presented poorer evaluations of their possibility to reach green spaces by walking (OR 1.94 (95% CI 1.14–3.32)) and the availability of relaxation areas (OR 2.30 (95% 1.34–3.95)). Other environmental variables did not show any significant relationships with diastolic blood pressure.

## 4. Discussion

Citizen Science, defined as participation of the general public in scientific research, is an opportunity for the citizens to familiarize themselves with scientific thinking and to raise societal awareness about the links between environmental issues and health. However, the engagement of volunteers that do not represent the population’s age structure and education, limits the generalization of the results of such studies [20]. The usage of standardized methods such as standard protocols, questionnaires adapted to the participant’s skills, and formulated research questions can ensure high levels of data quality. 

This citizen science research uses formalized questionnaires and standard protocols, which ensure data quality and helps to gain a better understanding of how the quality of the residential environment might affect physical activity, obesity, and hypertension. To our knowledge, this is the first environmental epidemiological study to investigate the associations between the participants’ concerns and perceptions of the quality of residence neighborhoods and health issues in an Eastern European country. There are some data indicating that in different European countries, area-level and individual-level socioeconomic characteristics operate differently [21]. We tested the hypothesis that the quality of physical and social neighborhood affects the citizens’ physical activity and is associated with obesity and blood pressure. Our hypothesis was partly confirmed. 

The findings of our study revealed that citizens highly scored the quality of their residential neighborhood, specifically noting that public transport met their needs in the district and that they had good opportunities for walking to reach the city’s green spaces or parks. The participants were mostly concerned about the harmful effect of air pollution and noise on health. The analysis of the perception of the residential environment by independent factors of the built environment showed some associations with the participants’ physical activity and other health issues. The participants’ rating scores of the pathways and cycling routes revealed concerns about the availability and maintenance of infrastructure that facilitates walking and cycling within the district and were related to physical activity, blood pressure, and the body mass index. These evaluated associations persisted after adjustment for sex, age, education level, smoking, and income.

The air quality concern and health problems as well as the perception of the effects of neighborhood characteristics differed across the districts at the spatial scale. Using traffic flows in the residence street to classify participants’ exposure level to traffic emissions, at the city level, we did not find significant evidence of the dependence of the air quality concern and the prevalence of hypertension (Figure 1). However, in two districts, we found a negative correlation. The comparison of the districts with a low prevalence of hypertension with those where the prevalence of hypertension was high showed that personal-level differences (body mass index, blood pressure, and chronic disease) were more responsible for a higher prevalence of hypertension in some districts than differences in environmental-level characteristics were. The participants of both groups reported low physical activity levels—only 11.8% of all the participants reached the overall physical activity level at of least 150 min/week of moderate-intensity physical activity outdoors by fast walking, biking, or gardening. This index indirectly shows a possibility to improve health status by increasing physical activity, since in the districts with a low prevalence of hypertension, when answering the question: “Are you satisfied with pathways and cycling routes?”, the participants presented high satisfaction scores (4.99 (0.11))—similarly to participants residing in districts with a high prevalence of hypertension (4.86 (0.15)). These data show that there is good infrastructure to undertake walking or cycling trips for physical activity; commuting and health-related behavior modifications might improve citizens’ health. There is a considerable need for increasing physical activity among Kaunas citizens. Based on the findings previous studies, even 30 min of supervised walking may have beneficial effects on the diastolic blood pressure, which can subsequently decrease other risk factors and improve health [11,22].

Our findings are consistent with the previously reported results suggesting that individual-level characteristics, physical activity, and residential characteristics may independently contribute to health outcomes such as blood pressure [6,23].

Previous research indicates the importance of environmental characteristics in promoting physical activity [24,25]. Areas with more walkable environments were characterized by higher rates of walking and cycling to work and school (i.e., active commuting) compared to areas with a less walkable environment [26]. However, only limited studies have examined the perceptions of the quality of the environment among citizens. Our findings are in line with the previously reported data indicating that the neighborhood and social environment may influence blood pressure, which is associated with modifiable and non-modifiable risk factors such as behavioral, social, and environmental risk factors that might produce stress and contribute to hypertension [27,28]. Blood pressure may be related to neighborhoods through multiple mechanisms. A potential chain includes the walking environment, the social environment, and chemical stress produced by air pollution. Evidence suggests that moderate physical activity and walking are effective measures to decrease blood pressure. Physical activity in green spaces might reduce stress levels and have a positive effect on lowering increased blood pressure.

The participants’ reports on physical activity levels showed a great variety of associations with the characteristics of the residential environment. The data of the multivariate logistic regression models (Table 4) revealed that citizens residing in neighborhoods with a better infrastructure for the possibility of reaching green spaces by walking, and the availability of relaxation areas had a lower probability of having an increased diastolic blood pressure than participants in worse neighborhoods did. Our data are in line with the reported findings indicating that residents of neighborhoods with better walkability, greater safety, and more social cohesion were less likely to be hypertensive [23].

Some evidence suggests that neighborhood characteristics were related to the body mass index, and physical activity might have a mediating effect on the association between residential environment and health outcomes [29]. The evaluation of the association between the quality of the neighborhood and obesity presented evidence that less physically active participants more often were obese and had a higher diastolic blood pressure. Obese participants more often were male, older, and less educated. We revealed an association between the scoring of the environmental quality and obesity. Obese participants presented significantly poorer scores of air pollution problems in the place of residence, poorer ratings of the quality of pathways and cycling routes, their possibility to reach green spaces by walking, and the availability of relaxation areas.

## 5. Strengths and Limitations of the Study

This citizen science study used an environmental epidemiological approach and presented evidence of associations between environmental quality ratings and health issues. The cross-sectional design of our study limits our ability to infer causal associations; however, multivariate logistic regression models of neighborhood quality and physical activity, obesity, and diastolic blood pressure revealed significant associations and evidence that an improving neighborhood has a potential to have a beneficial effect on citizens’ health. We were able to gather perceptions of environmental quality, characteristics of health concerns, and behavior at the individual and the district levels in a large sample of subjects. Variability due to individual subjectivity was averaged out by aggregating individual responses within a neighborhood, which increased the strength of the associations across areas. Moreover, we attempted to control for the key confounders such as sex, age, education level, smoking, and income. However, residual confounding by personal characteristics is possible. 

The limitations of the study also include self-reported health problems in response to a set of questions that have been found to be affected by measurement error as the result of recall bias associated with age and social disparity [30]. However, some studies have found good concordance between self-reported and more objective measures, such as medical records to identify disease history [31]. In this study, we used systematic methods such as designing standard protocols, questionnaires adapted to participants’ skills, and formulated research questions to ensure high levels of data quality. Moreover, the vast majority of the participants were of working age. The working-age participants had fever problems filling out the questionnaires and scoring the perceptions than the retired ones did, and that had a positive impact on the quality of the data such as accuracy in data collection by filling out questionnaires and scoring perceptions. In this study, we used subjective measures for physical activity characteristics, and therefore in future studies, integrating objective physical activity measurements by physical activity sensors would allow for achieving detailed situation data and increased sensitivity.

## 6. Conclusions 

Recent studies suggest that built residential characteristics have an effect on the physical activity and health of the citizens. However, the influence of the simultaneous evaluation of multiple build and social environmental characteristics and health issues has received less attention to date. The present study attempted to fill this gap by putting citizens’ concerns at the center of the environmental epidemiology research and developing a participatory citizen science project that would give answers to the citizens’ questions and would provide them with personalized information. We interacted with the citizens during face-to-face meetings to obtain information about their concerns and insights related to the environmental quality in the residential setting and health. The participants together with scientists designed the type of the study, the data to be collected, the procedures to collect the data, and the protocol. This study supports the current citizens’ research in environmental epidemiology, opening science to people from all backgrounds, and raising public awareness about the effects of urban pollution on health by translating scientific knowledge gained throughout the process into useful and practical knowledge for the society. Even though the present study did not employ a complete list of environment related disease risk factors, we studied the most important ones and found some of them to be associated with the characteristics of the built and social residential environment. 

The findings suggest that the neighborhood quality and individual-level characteristics were the factors that influenced a higher prevalence of health concerns at the district level. Both the environmental quality of the neighborhood and individual-level characteristics are important determinants of poor health and low physical activity, and may promote the development of obesity and hypertension. Our findings suggest that efforts to improve citizens’ well-being and health may benefit from attention to increasing physical activity and improving the physical and social environment of the neighborhood. 

## Figures and Tables

**Figure 1 ijerph-17-04420-f001:**
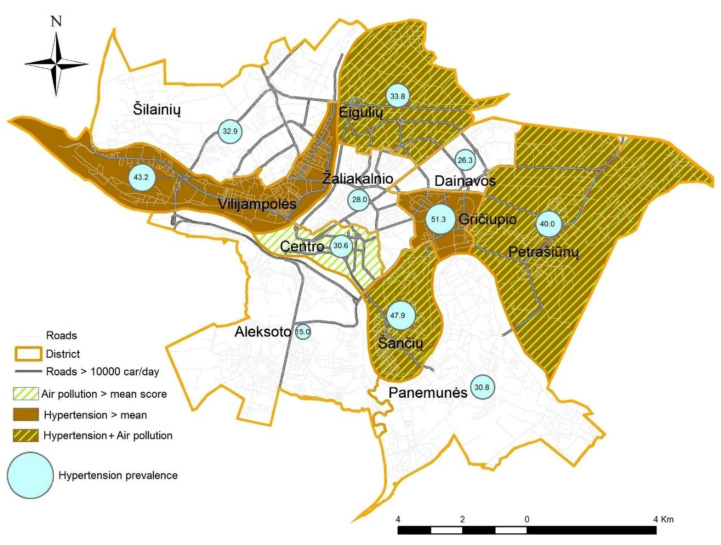
Spatial distribution of the perception of air pollution and the prevalence of hypertension.

**Table 1 ijerph-17-04420-t001:** Characteristics of the participants living in Kaunas city.

Characteristics		*n*	%
Age groups	18–44	359	61.7
	45–64	177	30.4
	≥ 65	46	7.9
Sex	Men	245	42.1
	Women	337	57.9
District	1	40	6.9
	2	49	8.4
	3	74	12.7
	4	95	16.3
	5	39	6.7
	6	13	2.2
	7	25	4.3
	8	48	8.2
	9	73	12.5
	10	44	7.6
	11	82	14.1
	Secondary	197	33.7
Education	College	129	22.2
	University	256	44.0
Family status	Married	305	52.4
	Other	277	47.6
Situation at work	Full-time	345	59.3
	Part-time	38	6.5
	Other	199	34.2
Monthly net income	Less than 400 Euro	98	16.8
	400–1000 Euro	484	83.2
Traffic 10,000 cars/day	No	454	78.0
	Yes	128	22.0
Duration of living in the place of residence, years (mean (SE))		18.0 (0.66)	

SE—standard error.

**Table 2 ijerph-17-04420-t002:** Self-reported physical activity and health measures in district groups by the prevalence of hypertension.

Variables	Districts with a Low Prevalence of Hypertension (< mean), *n* (%) or mean (SE)	Districts with a High Prevalence of Hypertension (> mean), *n* (%) or mean (SE)	*p*
Body mass index (BMI)	24.59 (0.25)	25.83 (0.35)	0.004 ^†^
BMI			0.026 ^‡^
<30	313 (62.4)	189 (37.6)	
BMI > =30 (obesity)	39 (48.8)	41 (51.2)	
Systolic blood pressure	124.58 (0.99)	129.90 (1.02)	<0.001 ^†^
Diastolic blood pressure	83.64 (0.70)	84.79 (0.87)	0.296 ^†^
Chronic disease			
No	218 (69.6)	95 (30.4)	<0.001 ^‡^
Yes	134 (49.8)	135 (50.2)	
Hypertension			<0.001 ^‡^
No	255 (65.7)	133 (34.3)	
Yes	97 (50.0)	97 (50.0)	
Health status			0.070 ^‡^
Good	309 (62.0)	189 (38.0)	
Poor	43 (51.2)	41 (48.8)	
Current smoking			0.141 ^‡^
No	236 (58.4)	168 (41.6)	
Yes	116 (65.2)	62 (34.8)	
Smoking duration	3.92 (0.45)	4.21 (0.59)	0.686 ^†^
Time outdoors			0.510 ^‡^
<150 min/week	308 (59.9)	206 (40.1)	
≥150 min/week	44 (64.7)	24 (35.3)	
Time in park (min/week)	22.95 (1.82)	24.13 (2.10)	0.678 ^†^

^†^*p* value of Student’s t test; ^‡^
*p* value of the chi-squared test; SE—standard error.

**Table 3 ijerph-17-04420-t003:** Mean ratings of the perceptions of neighborhood quality and social wellbeing in district groups by the prevalence of hypertension.

Questions	Districts with a Low Prevalence of Hypertension (< mean), mean (SE)	Districts with a High Prevalence of Hypertension (> mean), mean (SE)	*p*
Does the public transport in the district meet your needs?	5.39 (0.11)	5.20 (0.14)	0.267
Are you satisfied with pathways and cycling routes?	4.99 (0.11)	4.86 (0.15)	0.481
Are there opportunities for walking to reach the city’s green spaces or parks?	5.07 (0.12)	5.16 (0.14)	0.620
Do you regularly visit the natural environment?	4.23 (0.12)	4.10 (0.16)	0.520
Is there a place in your residential area adapted for exercise and relaxation?	4.56 (1.20	4.29 (0.16)	0.167
Does air pollution in your place of residence cause problems?	4.03 (1.12)	3.62 (0.14)	0.027
Does the noise in your place of residence hinder your sleep and/or work at home?	4.76 (0.14)	4.75 (0.14)	0.973
Are there public spaces and rooms to meet people available in your residential area?	4.09 (0.20)	3.90 (0.15)	0.300
Do you feel safe in your area?	5.24 (0.10)	4.97 (0.14)	0.113
Can you take part in decision-making to improve the environment in which you live?	3.19 (0.12)	3.46 (0.16)	0.178
During the last 6 months, have you felt stress, tension, or anxiety?	4.26 (0.11)	4.13 (0.14)	0.468

All neighborhood perception scores ranged from 1 to 7: 1 = strongly disagree, and 7 = strongly agree. Higher scores indicate better neighborhood conditions.

**Table 4 ijerph-17-04420-t004:** Multivariate logistic regression models of neighborhood quality items on physical activity, obesity, and diastolic blood pressure.

Independent Variables	Dependent Variable Models					
	Physical Activity (more 150 min/week)		Obesity (BMI = >30)		Diastolic Blood Pressure (≥ 90)	
	Crude OR (95% CI)	Adjusted OR * (95% CI)	Crude OR (95% CI)	Adjusted OR * (95% CI)	Crude OR (95% CI)	Adjusted OR * (95% CI)
Quality of pathways and cycling routes						
<mean score	1.70 (1.02–2.83)	1.70 (1.02–2.85)	2.25 (1.40–3.62)	1.95 (1.17–3.27)	1.77 (1.07–2.94)	1.51 (0.89–2.59)
Green spaces by walking						
<mean score	0.70 (0.42–1.18)	0.68 (0.40–1.15)	1.72 (1.07–2.77)	1.62 (0.97–2.72)	2.12 (1.27–3.54)	1.94 (1.14–3.32)
Regular visits to green spaces						
<mean score	0.41 (0.24–0.70)	0.40 (0.24–0.70)	1.25 (0.78–2.02)	1.08 (0.64–1.82)	1.39 (0.84–2.31)	1.20 (0.71–2.04)
Available relaxation area						
<mean score	0.71 (0.42–1.19)	0.70 (0.42–1.18)	2.50 (1.53–4.09)	2.37 (1.39–4.02)	2.46 (1.47–4.13)	2.30 (1.34–3.95)
Air pollution problems						
<mean score	0.86 (0.51–1.43)	0.87 (0.52–1.47)	2.07 (1.28–3.45)	1.87 (1.11–3.15)	1.11 (0.67–1.83)	1.02 (0.60–1.74)
Noise problems						
<mean score	0.37 (0.20–0.68)	0.37 (0.20–0.67)	1.30 (0.81–2.09)	1.05 (0.63–1.77)	1.30 (0.79–2.14)	1.14 (0.67–1.94)
Safety						
<mean score	0.76 (0.45–1.26)	0.77 (0.46–1.28)	1.12 (0.70–1.80)	1.18 (0.71–1.96)	0.88 (0.54–1.46)	0.90 (0.53–1.53)
Traffic 10,000 cars/day						
No	1	1	1	1	1	1
Yes	1.21 (0.67–2.18)	1.21 (0.67–2.18)	0.53 (0.27–1.02)	0.39 (0.19–0.82)	1.02 (0.56–1.88)	1.08 (0.57–2.05)

OR, odds ratios; * adjusted for: sex, education level, age, smoking status, and income; Neighborhood quality scales ranged from 1 to 7. For all scales, the referent group is the mean score or above.

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
