# Peer review of "Environmental Quality Perceptions and Health: A Cross-Sectional Study of Citizens of Kaunas, Lithuania"

_ijerph, 2020, doi:10.3390/ijerph17124420_

Round 1

Reviewer 1 Report

This study examined the cross-sectional association between perceived urban environments and people’s health-related characteristics. The study design is valid. However, the manuscript is not presented in good quality. Specific comments:
L80. Why recruit people from the conference? What is the type of conference? Conference attendees may share some characteristics, such as occupations.
L100-L101. Is the “Most interested” based on survey results?
L102. Please specify the name of the software.
L153. Please clarify the cutoff value for “higher prevalence” and “lower prevalence”. Although it has been mentioned in the following manuscript, but it would be best if you mentioned it for the first time it appears in the manuscript.
L173. How to find out people exposed to heavy traffic emission? What is the standard for “exposed to heavy traffic emission”?
Table 3. The last question, “During the last 6 months, have you felt stress, tension, or anxiety?” seems discordant from all the other questions.
L259. My question here is similar to the question of Table 3. Research through the entire manuscript, it is not clear that what study aim would be. It seems that the author wants to link the environmental factors (built environment and air pollution) to people’s health. However, the manuscript also mentioned physical activity and obese, or exposure of area with a high prevalence of hypertension. I would suggest focusing on the exposure of perceived built environment and air pollution and the health of study participants.

Reviewer 2 Report

The paper describes an attempt to estimate statistic correlation between characteristics of the build environment, citizen health and their perception of quality of life.

The paper is somewhat double faceted. It is a case study (namely upon the city of Kaunas) but the authors also propose a general approach and generalized conclusions. The paper shows undoubtedly a high level for what concerns the screening activities, although a general objection will be outlined, but discussion and, expecilly, conclusions seem not adequate. Some points follow.

English is good in general but I would check for verbosities and lack of punctuation. Thus I would make a further check but I am not mother tongue. Figures are ok while tables badly need a general reformatting.

Going to the paper content, I was surprised to see the composition of participants. The authors write “The participants of this study were 576 citizens; the majority of them (61.7%) were 18-44 years 170 of age, and 44% of all the participants had university degrees. Are the authors persuaded that this can be considered a truly indicative sample of Kaunas population? In my mind there is an excess of young people and graduated individuals. The authors should discuss in deep this point and add proper link in discussion, conclusions and, possibly, within limitations.

Another issue is that the authors show the results of their statistical analysis but they do not introduce properly such statistic models and instruments. I think, just for instance, that simply talking of p-values without any explanation, could lead to misinterpretation. By the way, at page4, the authors say “Predictor variables whose univariate tests showed an association of p<0.2 (Field, 2017; Hosmer et al., 2013) with the outcome or those that changed the adjusted odds ratios (aOR) by 10% or more were retained for inclusion in the multivariate logistic regression analysis”. Although they properly refer to Field and Hosmer, are we sure that p-values of 0.2 reveal a proper significance? Also, tables show a number of very high p-values that should be better addressed and explained. Finally, the authors should confirm that p-values-0.000 mean smaller than…

Finally I would ask the authors to enhance conclusions.

Round 2

Reviewer 1 Report

I have no additional comments. But I think this manuscript still needs a extensive English editing.

Author Response

Dear Reviewer,

We thank  for your suggestions how the article could be improved.

The manuscript has been revised taking into account to all the Reviewers comments.

The revised manuscript ID: ijerph-789892 of 1st June, 2020, repeatedly underwent extensive English editing using a professional English editing service.

Corresponding author                                                                                         Regina Grazuleviciene

Reviewer 2 Report

The authors have addressed all the open points.

Author Response

Dear Reviewer,

We thank the Reviewer for their suggestions how the article could be improved.

The manuscript has been revised taking into account to all the Reviewers comments.

The revised manuscript ID: ijerph-789892 of 1st June, 2020, repeatedly underwent extensive English editing using a professional English editing service.